# Numerical Analysis on Global Serviceability Behaviours of Tall CLT Buildings to the Eurocodes and UK National Annexes

**Xuan Zhao, Binsheng Zhang \*, Tony Kilpatrick and Iain Sanderson**

Department of Civil Engineering and Environmental Management, School of Computing, Engineering and Built Environment, Glasgow Caledonian University, Glasgow G4 0BA, Scotland, UK; Xuan.Zhao@gcu.ac.uk (X.Z.); A.R.Kilpatrick@gcu.ac.uk (T.K.); I.Sanderson@gcu.ac.uk (I.S.)
\* Correspondence: Ben.Zhang@gcu.ac.uk; Tel.: +44-141-331-8660

**Abstract:** Cross-laminated timber (CLT) is an innovative engineered timber product and has been widely used for constructing tall timber buildings due to its excellent structural performance and good strength with its multi-layers of boards in both perpendicular directions. However, the global serviceability performance of tall timber buildings constructed from CLT products for the lift core, walls, and floors under wind load is not well known yet, even though it is crucial in a design. In this study, the finite element software SAP2000 is used to numerically simulate the global static and dynamic serviceability behaviours of a 30-storey tall CLT building assumed in Glasgow, Scotland, UK. The maximum horizontal storey displacement due to wind is only 16.6% of the design limit and the maximum global horizontal displacement is only 13.8% of the limit set to the Eurocodes. The first three lowest vibrational frequencies, modes and shapes were obtained, with the fundamental frequency being 19.9% larger than the code-recommended value. Accordingly, the peak acceleration of the building due to wind was determined as per the Eurocodes and ISO standard. The results show that the global serviceability behaviours of the building satisfy the requirements of the Eurocodes and other design standards. Parametric studies on the peak accelerations of the tall CLT building were also conducted by varying the timber material properties and building masses. By increasing the timber grade for CLT members, the generalised building mass and the generalised building stiffness can all be adopted to lower the peak accelerations at the top level of the building, so as to reduce human perceptions of the wind-induced vibrations with respect to the peak acceleration.

**Keywords:** tall timber building; cross-laminated timber (CLT); wind load; serviceability; storey and global horizontal displacements; vibrational frequency and mode; peak acceleration

## 1. Introduction

Modern novel engineered timber products have been expansively developed, and pre-fabricated timber components and prebuilt composite building modules provide many potential economic benefits. Hence, over the past decade, these timber products and components have become popular for constructing tall timber buildings in cities all around the world. As a renewable and sustainable construction material, timber can provide carbon sequestration and has much less embodied energy. The technical document Timber as a Sustainable Building Material [1] comprehensively compared construction materials such as timber, steel, concrete, aluminium and brick with respect to fossil fuel energy, $CO_2$ release and storage, energy requirement, embodied energy, environmental externality cost, etc. Based on the data from the technical reports Environmental Properties of Timber by Ferguson et al. [2] and Building Materials Energy and the Environment by Lawson [3], timber materials release less greenhouse gases and wastes and also can be recycled conveniently or reused as fuel when compared with other construction materials. Hart and Pomponi [4] regarded timber as the lower carbon option without considering the carbon storage.

Due to the environmental advantages of timber construction, extensive research and development have been conducted on medium-rise timber buildings, and now high-rise

timber buildings have attracted the attention of architects and design engineers. Preliminary investigations into multi-storey buildings by Polastri et al. [5–8] indicated that cross-laminated timber (CLT) cores could be feasibly used for tall and slender timber buildings. For tall timber buildings, popular structural systems include the CLT shear wall structural system, glued laminated timber (Glulam) frame structural system, and timber–concrete composite (TCC) structural system. Recently, the CLT shear wall structural system has become a popular option for designing and constructing tall timber buildings. Table 1 illustrates tall timber buildings over eight storeys that have been recently completed in the world.

**Table 1.** Recently completed tall timber buildings over eight storeys in the world.

| No | Building Name | Location | Storeys | Completed Year |
|----|---------------|----------|---------|----------------|
| 1 | Mjosa Tower | Brumunddal, Norway | 18 | 2019 |
| 2 | HoHo Tower | Vienna, Austria | 24 | 2019 |
| 3 | Brock Commons | Vancouver, Canada | 18 | 2017 |
| 4 | Banyan Wharf | London, UK | 10 | 2015 |
| 5 | Puukuokka | Jyväskylä, Finland | 8 | 2015 |
| 6 | Treet | Bergen, Norway | 14 | 2015 |
| 7 | Contralaminada | Lleida, Spain | 8 | 2014 |
| 8 | Strandparken | Stockholm, Sweden | 8 | 2014 |
| 9 | St. Dié-des-Vosges | Vosges, France | 8 | 2014 |
| 10 | Cenni di Cambiamento | Milan, Italy | 9 | 2013 |
| 11 | Pentagon II | Oslo, Norway | 8 | 2013 |
| 12 | Wagramer Strasse | Vienna, Austria | 10 | 2013 |
| 13 | Panorama Giustinelli | Trieste, Italy | 10 | 2013 |
| 14 | Maison de l'Inde | Paris, France | 10 | 2013 |
| 15 | Forte | Melbourne, Australia | 10 | 2012 |
| 16 | Lifecycle Tower One | Dornbirn, Austria | 8 | 2012 |
| 17 | Holz 8 | Bad Aibling, Germany | 8 | 2011 |
| 18 | Bridport House | London, UK | 8 | 2010 |
| 19 | Stadthaus | London, UK | 9 | 2009 |

With the development of modern construction technology for timber buildings, the heights of the buildings are also increasing. The third tallest building in the world is Brock Commons Tallwood House, an 18-storey timber student residence hall with a total height of 54 m at the University of British Columbia (UBC) in Vancouver, Canada, and was completed in 2017 [9]. The whole building consumed a total of 2233 m$^3$ CLT, Glulam, and parallel strand lumber (PSL), which could reduce $CO_2$ emission by 2432 tonnes. Extensive fire protection strategies were adopted for this building to gain a 2-h fire resistance. During the construction stage, multiple layers of Type X gypsum boards gave a 120-min fire resistance for all mass timber elements. During the completion stage, both active and passive fire protection strategies were utilised [10]. The second tallest building in the world is the HoHo Tower in Vienna, Austria. It is a 24-storey timber tower with a height of 84 m. A total of 74% of the building was made of wood and this can reduce $CO_2$ emission by 2800 tonnes [11]. The tallest timber building in the world is the Mjøsa Tower in Brumunddal, Norway. It is an 18-storey timber building, with a total height of 85.4 m [12]. The illustrative photos of these three tallest timber building in the world can be found from the given references [9–12].

Compared with concrete and steel structures, timber structures have a lower stiffness and mass density. This means that it is necessary to consider the horizontal sway and dynamic behaviour when designing tall timber structures. To increase the dynamic performance, Szczepańsk et al. conducted dynamic tests and recognised that some thermal insulation materials could enhance the stiffness and damping properties [13]. The base shear forces on tall timber buildings due to wind loading are normally larger than those due to earthquake loading because of the high flexibility of these buildings [14]. During the structural design of tall timber buildings, it is necessary to consider the effects caused by wind loading under the ultimate limit states and check the wind-induced lateral displace-

ments under the serviceability limit states. The design limits for wind-induced horizontal deflections are not provided in the current structural Eurocodes. It is generally accepted that a limit of $H/500$ is appropriate for wind-induced horizontal deflections of each storey in a multi-storey building, or the structure as a whole for a multi-storey building where $H$ is the storey height or the total building height [15]. Through experimental investigations, Liang et al. indicated that the displacement is not the only reliable indicator under seismic performance for timber frame structures [16]. Human response to wind-induced vibrations in tall timber buildings is also important. The human response is difficult to measure, so the most common way to assess wind-induced vibrations in buildings is to evaluate the accelerations at the top of these buildings and the frequencies or periods of the lower vibration modes, normally the lowest three modes. In this research, extensive investigations into the relationships between timber grade, mass, stiffness, and peak acceleration for a tall CLT timber building are conducted.

## 2. Basic Plans of the Tall CLT Building

The tall CLT building that was studied is a 30-storey office building that mainly consists of cross-laminated timber (CLT) elements. It was designed and analysed under ultimate limit states (ULS) and serviceability limit states (SLS) according to the relevant structural Eurocode parts and the corresponding UK National Annexes (UK NA), under normal temperature and fire conditions. This building is a design project and for convenience was assumed to be located in Glasgow, Scotland, UK. Figure 1a illustrates the plan view of the building, and the floor plan is the same for all storeys except the thickness of the vertical CLT elements. The storey height of the CLT building is 3.5 m, giving the total height of the building as 105 m. The timber used for manufacturing the CLT elements is grade C24 softwood, whose strength and stiffness properties are given in BS EN 338 [17]. The length and width of the building are 39 m and 34 m, respectively. The front and side views of the building are also illustrated in Figure 1b,c, where the red rectangular box indicates Wall 1, which is the 360-mm-thick 9-layer uniform CLT inner core. The blue lines on the red line indicate that there are no huge openings on the core wall. The whole tall CLT building is designed as a structure with an inner CLT core and CLT shear walls. The grey lines indicate the CLT shear walls for the building, where Wall 2 represent the 360-mm-thick 9-layer CLT walls for Storeys 1 to 15, and Wall 3 represent the 280-mm-thick 7-layer CLT walls further up. All floor slabs for this building were constructed from 245-mm-thick 7-player CLT panels. The details of the different CLT elements are listed in Table 2.

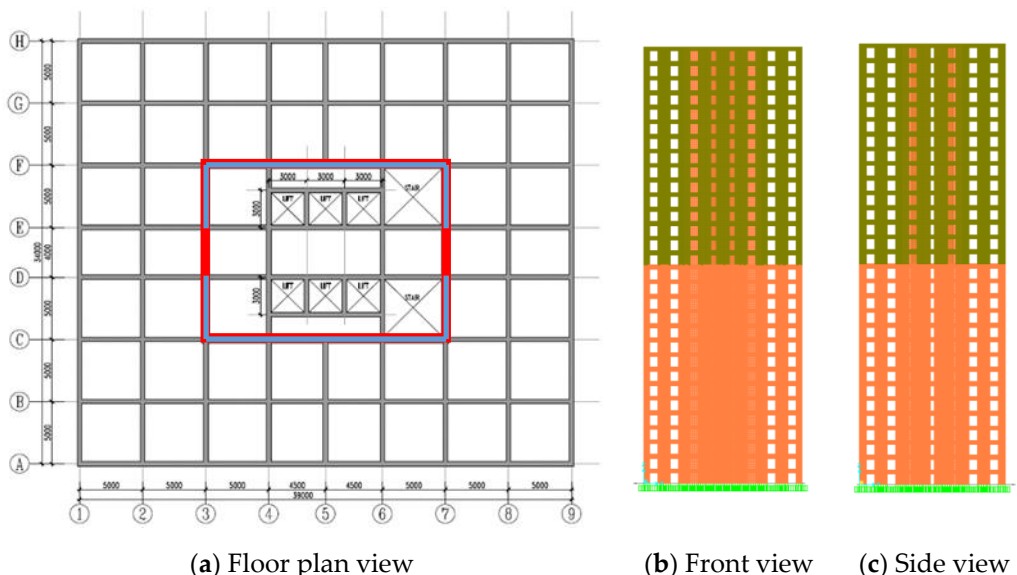

(**a**) Floor plan view      (**b**) Front view    (**c**) Side view

**Figure 1.** Different views of the 30-storey-tall cross-laminated timber (CLT) building.

**Table 2.** Details of the CLT elements for the studied tall CLT building.

| CLT Elements | Number of Layers | Layer Thickness $h_i$ | Overall Thickness $h_T$ |
|---|---|---|---|
| Wall 1 (red box in Figure 1) | 9 | 40 mm | 360 mm |
| Wall 2 (grey lines in Figure 1) between Storeys 1 and 15 | 9 | 40 mm | 360 mm |
| Wall 3 (grey lines in Figure 1) between Storeys 16 and 30 | 7 | 40 mm | 280 mm |
| Floor slabs | 7 | 35 mm | 245 mm |

As shown in Figure 1, with the front and side views of the building, most CLT wall elements have openings of minimum 1.0 m × 2.5 m for doors or windows, except part of the core shown by the blue lines. The dimensions of the door openings on various CLT wall elements are listed in Table 3, which were selected to ensure easy access. The selected opening dimensions also need to ensure that all the remaining CLT elements (excluding connections) are strong and stiff enough to withstand the design loads under ULS and SLS and have a 2-h fire-resistance under standard fire conditions.

**Table 3.** Sizes of the wall openings.

| Wall Width | Sizes of the Wall Openings (Width × Height) |
|---|---|
| 5.0 m | 2.0 m × 2.5 m |
| 4.5 m | 1.5 m × 2.5 m |
| 4.0 m | 1.0 m × 2.5 m |

Because of the memory capacity of the finite element analysis software and the complexity of the connections, connections were not simulated in this research. The connections were assumed to be fully protected with plasterboards or wood panels and the structural effects of the actual connections were not considered here. In this study, all the material properties of the C24 softwood timber for producing CLT elements were directly quoted from BS EN 338 while the characteristic rolling shear strength was quoted from ETA-06/0138 [18].

For the CLT walls, both sides are to be covered with two layers of 15-mm-thick type F plasterboards. For the CLT floor slabs, only one side is to be covered with two layers of 15-mm-thick type F plasterboards as ceilings.

## 3. Finite Element Analysis (FEA)

To numerically simulate the static and dynamic performances of the studied tall CLT building and conduct a structural analysis and design of the building subjected to lateral wind loading, it is necessary to use the finite element analysis technique. The static performance of individual CLT elements could be verified according to the pre-established design procedures [19]. However, due to the complex geometric configurations of the building, numerical simulations are the better option than traditional analytical methods for assessing its dynamic performance. The finite element method (FEM) is a numerical technique used to perform numerical analysis based on given physical laws. In this study, the commercial finite element software SAP2000 [20] is used as a tool to investigate the static behaviours and dynamic performances of different structural systems for the tall CLT building under various loading schemes. SAP2000 is also used to numerically simulate various types of building models by applying different assumptions of material properties and boundary constraints. The aspects to be used in preparation for the numerical modelling include the units and grids of the models, the adopted materials, sections, constraints, load patterns, load cases, and load combinations for design purposes, which should all be defined by the user before starting the modelling. More details can be found from the SAP2000 manual [21].

### 3.1. The Estimation Method

To reduce the large data caused by small element sizes and multi-layered materials, the estimation method is used to simplify and convert multi-layer elements into single-layer thin orthotropic shell elements for finite element analysis modelling because CLT is a multi-layer orthotropic material [22,23]. The properness of the estimation method was then confirmed by the experimental projects conducted by the Ministry of Land, Infrastructure, Transport and Tourism (MLIT) and the Forestry Agency in Japan [23]. For a more accurate analysis, however, this method turned to consider the contributions of perpendicular layers to the grain. In the following numerical models, the CLT panels are modelled as single-layer orthotropic shell elements. The directions of the CLT panels are defined in Figure 2.

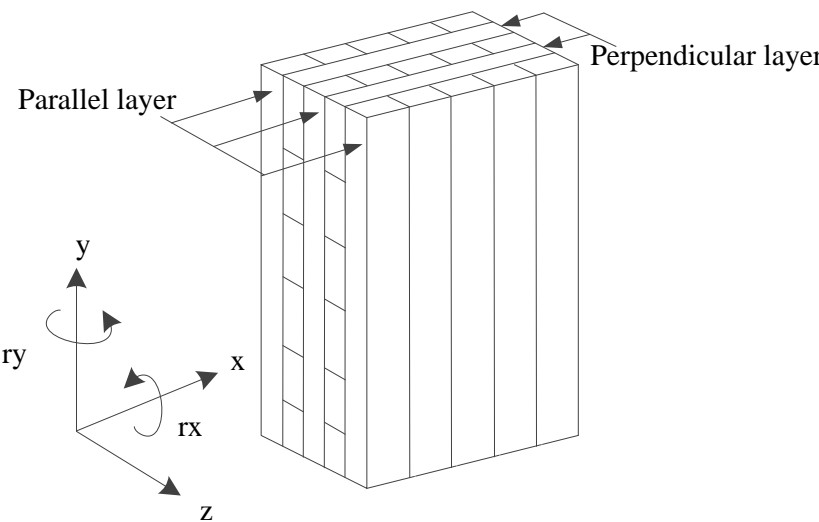

**Figure 2.** Definition of the directions of the CLT panels.

### 3.1.1. Out-of-Plane Elastic and Shear Moduli

The equations for determining the out-of-plane elastic bending and shear moduli of the CLT panels are given in the CLT Handbook [24]. Figure 3 illustrates a typical 5-layer CLT element.

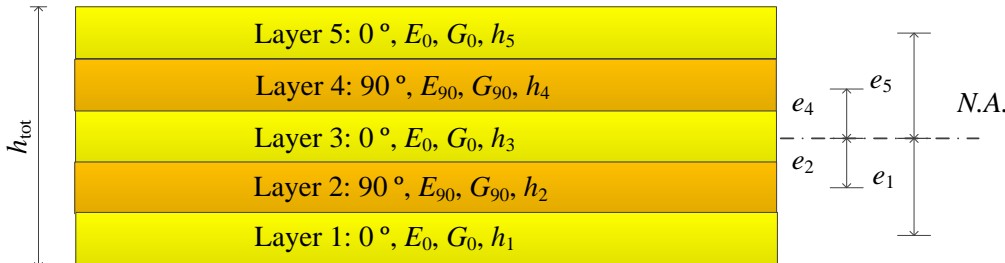

**Figure 3.** Typical 5-layer CLT element.

The equivalent plate bending stiffness $K_{\text{CLT}}$ can be analysed as

$$K_{\text{CLT}} = \sum_{i=1}^{n} E_i \, I_i + \sum_{i=1}^{n} E_i \, A_i \, e_i^2 = \sum_{i=1}^{n} E_i \left( I_i + A_i \, e_i^2 \right) \tag{1}$$

where $E_i$ is the elastic modulus of layer i, $I_i$ is the moment of inertia for the cross-section of layer i, $A_i$ is the cross-sectional area for layer i, and $e_i$ is the distance between the centre of layer i and the global neutral axis.

The equivalent shear stiffness $S_{CLT}$ can be analysed as

$$S_{CLT} = \frac{a^2}{\left(\frac{h_1}{2G_1\,b}\right) + \left(\sum_{i=2}^{n-1}\frac{h_i}{G_i\,b_i}\right) + \left(\frac{h_n}{2\,G_n\,b_n}\right)} \quad (2)$$

where $b_i$ and $h_i$ are the width and thickness of layer i, $a$ is a geometric parameter and $a = h_{total} - h_1/2 - h_n/2$, and $G_i$ is the shear modulus of layer i.

Though the estimation method only considers the parallel layers, the elastic moduli and strengths of the cross layers are considered in this study. The contributions of the cross layers can be regarded as those of parallel layers but with reduced effects. Some equations can be upgraded for use.

For out-of-plane bending, the equivalent bending elastic moduli about the x and y axes are termed $E_{rx}$ and $E_{ry}$. The equivalent shear modulus in the zx-plane is termed $G_{zx}$ and the shear modulus in the yz-plane is termed $G_{yz}$. These elastic moduli can be determined from the following equations:

$$E_{rx} = \frac{K_{CLT,rx}}{I_{full,x}} \text{ and } E_{ry} = \frac{K_{CLT,ry}}{I_{full,y}} \quad (3)$$

$$G_{zx} = \frac{S_{CLT,ry}}{A_{full,y}} \text{ and } G_{yz} = \frac{S_{CLT,rx}}{A_{full,x}} \quad (4)$$

### 3.1.2. In-Plane Elastic and Shear Moduli

For in-plane bending, the normal equivalent elastic moduli about the x and y axes are termed $E_x$ and $E_y$, with the shear modulus in the xy-plane termed $G_{xy}$. These elastic moduli can be determined from the following equations:

$$E_x = \frac{\sum_{i=1}^{n} E_{x,i} h_{x,i,in}}{h_{tot}} \text{ and } E_y = \frac{\sum_{i=1}^{n} E_{y,i} h_{y,i,in}}{h_{tot}} \quad (5)$$

$$G_{xy} = \frac{E_y}{e_1} \quad (6)$$

where $E_{x,i}$ and $E_{y,i}$ are the elastic moduli of layer i about the x and y axes, $h_{x,i,in}$ is the transformed thickness of layer i in the x direction for in-plane bending, $h_{y,i,in}$ is the transformed thickness of layer i in the y direction for in-plane bending, $h_{tot}$ is the original total thickness of the CLT panel, $e_1$ is the ratio of the elastic modulus to the shear modulus in the longitudinal direction, and n is the total number of layers.

### 3.2. Setup of the Numerical Model in SAP2000

The effects of the actual connections have not been considered yet and all the CLT elements were assumed to be rigidly connected with fixed boundary conditions. In this research, all the CLT elements were modified as 2D thin-shell elements with a mesh size of 500 mm × 500 mm. The materials of the shell elements are regarded as orthotropic materials. In-plane stiffnesses were applied to all CLT wall elements while out-of-plane stiffnesses were applied to all CLT floor slab elements.

All the effective material properties were obtained using the estimate method and then input into SAP2000. All the structural CLT panels were defined as orthotropic thin-shell elements. Rigid diaphragms were assumed for all floor slabs. The wind load was defined as exposure from the extents of rigid diaphragms in a wind direction of 270°. The wind velocity was 26.21 m/s, the terrain category set as II, and the orography factor, turbulence factor, and structural factor were all set as 1.0. The applied dead load was 0.92 kN/m² on the floor slabs, excluding the self-weight, and the applied imposed load was 3.3 kN/m². The extra dead load caused by the plasterboards, connections, and decorations on the walls

was assumed to be 5 kN/m. As shown in Figure 1a, there are some areas used as stairs. To assemble the model, the thickness of these stair areas was adopted as 0 mm, with an assumed dead load of 4 kN/m² and assumed imposed load of 3 kN/m². The actions on the building are chosen from BS EN 1991-1-1 [25] and the corresponding UK NA [26]. The load combinations were based on BS EN 1990 [27] and the corresponding UK NA [28]. The utilised stiffness properties of the individual CLT elements are listed in Table 4.

**Table 4.** Equivalent stiffness parameters of the CLT elements.

| Stiffness Parameters | Walls 1 and 2 | Wall 3 | Floor Slab |
|:---:|:---:|:---:|:---:|
| $t$ | 360.0 mm | 280.0 mm | 245.0 mm |
| $E_{rx}$ | 7441 N/mm² | 7933 N/mm² | 7929 N/mm² |
| $E_{ry}$ | 3927 N/mm² | 3438 N/mm² | 3436 N/mm² |
| $E_x$ | 5094 N/mm² | 4926 N/mm² | 4926 N/mm² |
| $E_y$ | 6276 N/mm² | 6444 N/mm² | 6444 N/mm² |
| $G_{xy}$ | 392.22 N/mm² | 402.77 N/mm² | 402.77 N/mm² |
| $G_{yz}$ | 111.53 N/mm² | 107.54 N/mm² | 107.55 N/mm² |
| $G_{zx}$ | 111.53 N/mm² | 107.54 N/mm² | 107.55 N/mm² |

To simplify the numerical simulations of the CLT elements, the Poisson ratio $v$ for all CLT materials was assumed to be zero to account for cracking parallel to the grain in a lamination layer or for the dry joints when no edge gluing is applied. Actually, most CLT elements are edge glued, but the glues have no technical approvals for load-bearing purposes. Hence, the joints have to be assumed dry as if there are no glues [29]. Section 5.1.1 in BS EN 1995-2 [30] also specifies that Poisson's ratio $v$ may be taken as zero when analysing the deck plates made of softwood laminations. Single CLT members can be verified to the Canadian CLT Handbook and Wallner-Novak's book [31], and other timber members and connections can be verified to BS EN 1995-1-1 [32] and the corresponding UK NA [33].

## 4. Global Structural Performances of the Tall CLT Building

### 4.1. Vibrational Frequencies, Modes, and Shapes

In the whole project, two different tall timber buildings were analysed. Model A refers to the tall CLT building, while Model B refers to the tall Glulam building. In this research, only the results for the tall CLT building are presented. After running the dynamic analysis on the 30-storey CLT building model (Model A), the modal frequencies and shapes for the first three lowest modes were obtained and are illustrated in Figure 4.

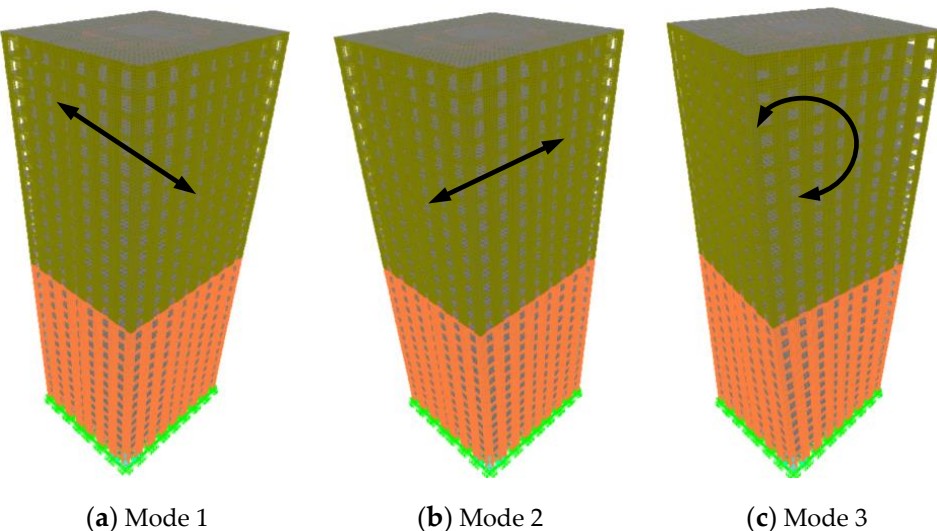

(**a**) Mode 1　　　　　　(**b**) Mode 2　　　　　　(**c**) Mode 3

**Figure 4.** The first three vibrational modes of Model A for the studied building.

As show in Figure 4, the first vibrational mode occurs along the weak y direction, with a modal frequency of 0.525 Hz. The second vibrational mode occurs along the strong x direction, with a modal frequency of 0.579 Hz. The third vibrational mode is the rotation mode about the vertical z-axis, with a modal frequency of 0.678 Hz. The first three lowest vibrational modes of the studied building are the primarily considered ones for structural engineers to check when designing tall reinforced concrete or steel buildings. In this study, the same three lowest vibrational modal characteristics, including the modal frequencies and shapes, are considered for analysing tall timber buildings.

From BS EN 1991-1-4 [34], the fundamental flexural frequency of a multi-storey building, $f_1$, for a height greater than 50 m can be estimated as

$$f_1 = \frac{46}{h} \tag{7}$$

This formula is obtained from the analyses of extensive experimental results on tall reinforced concrete and steel building structures. Based on this, the frequency of the currently modelled tall CLT building should be estimated as $n = 46/105 = 0.438$ Hz. In this study, the lowest modal frequency of the studied building is numerically obtained as 0.525 Hz, which is 19.9% larger than the code-recommended value. A number of factors may contribute to the difference in the fundamental frequency, which is associated with the stiffness and mass of the building. Model A contains many pre-assumed door openings on the CLT walls and the inner CLT lift core. There is one opening on most walls, in particular on the outer CLT core. Model A is also assumed to have rigid connections between the CLT elements by disregarding the effects of the connections. Reynolds et al. [35] suggested that a formula $f_1 = 55/h$ is a more accurate estimation for multi-storey timber buildings over 50 m. From this, the frequency is estimated as $f_1 = 55/105 = 0.524$ Hz, which is almost the same as the numerically obtained frequency. Hence, the obtained first frequency of 0.525 Hz in the weak y direction is fairly accurate and convincing.

### 4.2. Static Storey and Global Horizontal Displacements

The horizontal displacements at various building heights of Model A for the studied building caused by the wind load are illustrated in Figure 5. The maximum horizontal displacement of the building occurs at the top level of the building, with a value of 28.90 mm. It is generally accepted that the limit for the maximum horizontal displacements of buildings is $h/500$ or 210 mm in this study. The numerically determined displacement for the studied building reaches only 13.8% of the limit in accordance to the structural Eurocodes. Because the building model does not include the effects of connections between the CLT elements, the actual displacements are expected to be larger than the currently estimated values. SAP2000 also does not consider the contribution of shear deformations.

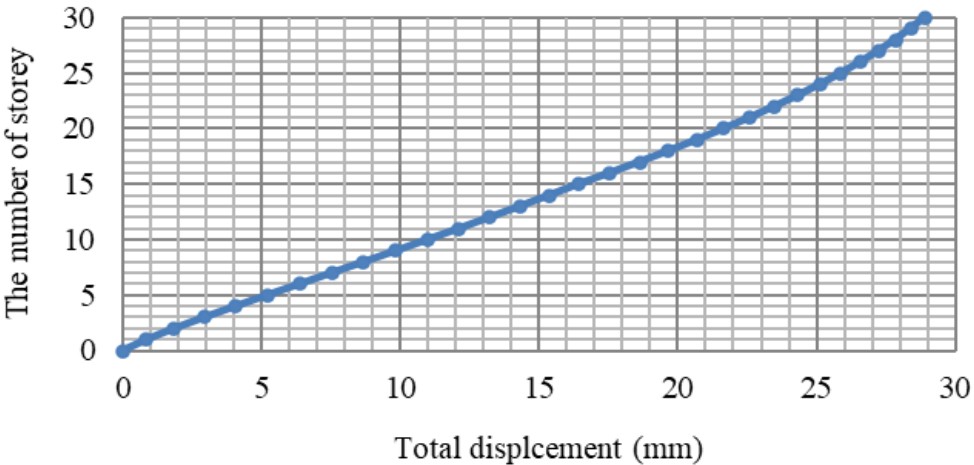

**Figure 5.** Horizontal displacements at various storey heights of the studied building.

Figure 6 illustrates the horizontal storey drifts of the studied building. The maximum storey drift occurs for Storey 7 at 1.162 mm, which is only 16.6% of the storey drift limit given as 3500/500 = 7 mm.

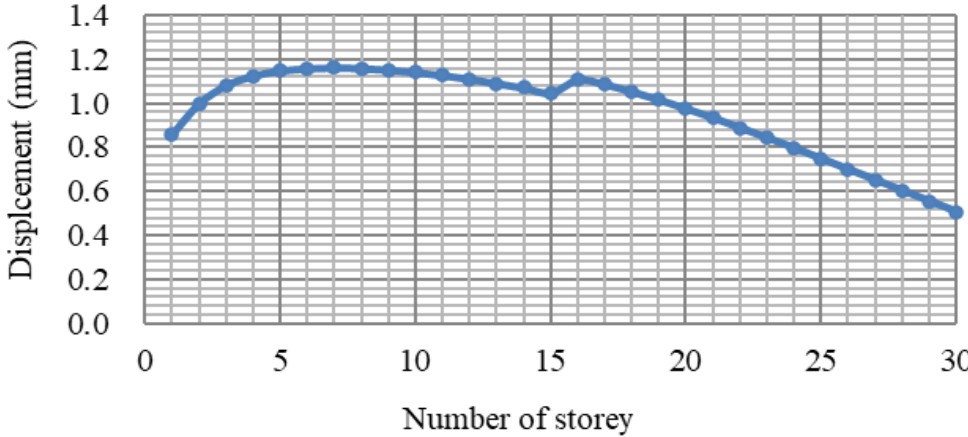

**Figure 6.** Storey drifts of the studied building due to wind load.

## 5. Peak Acceleration at the Top Level of the Tall CLT Building Due to Wind Load

When designing a timber structure for the serviceability limit states to Eurocode 5, the structural Eurocodes do not provide guidance or regulations about the effects of wind-induced vibrations and horizontal acceleration. For residential buildings, ISO 10137 [36] states that the accelerations should be kept within the limits of the daily living conditions with respect to human response to relatively ordinary motions of buildings, and horizontal accelerations of building with a one year-return period [37]. In Annex D of ISO 10137, which gives the guidelines for human response to wind-induced horizontal motions in buildings, horizontal accelerations due to wind loading with a return period of one year are applied. Figure 7 illustrates the criteria based on the peak accelerations corresponding to the fundamental natural frequencies of buildings in the principal translational and torsional directions.

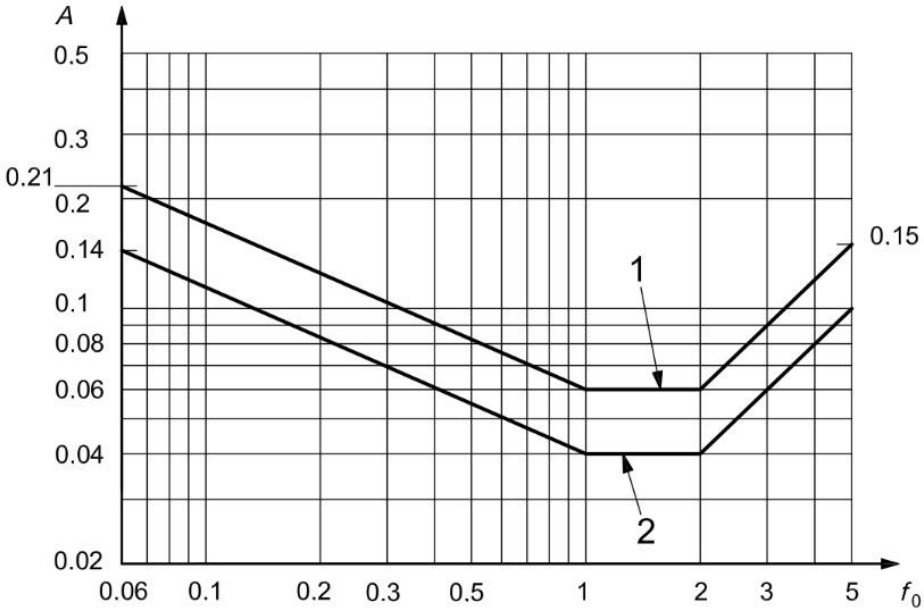

**Figure 7.** Evaluation curves for wind-induced vibrations in buildings along a horizontal (x,y) direction for a one-year return period. Key: $A$—peak acceleration in m/s$^2$; $f_0$—first natural frequency in a structural direction of a building and in torsion in Hz; 1—offices; 2—residences.

### 5.1. Manual Analysis of the Peak Acceleration at the Top Level of the Studied Building

BS EN 1991-1-4, the corresponding UK NA [38], and ISO 10137 are the main design standards used in this research. The horizontal peak acceleration of the building structure, $A(z)$, is calculated as

$$A(z) = k_\mathrm{p}\, \sigma_{\mathrm{a,x}}(z) \tag{8}$$

where $k_\mathrm{p}$ is a peak factor. The term $\sigma_{\mathrm{a,x}}$ is the standard deviation of the characteristic along-wind acceleration of the structural point at height $z$ and is obtained using Equation (B.10) of BS EN 1991-1-4 as

$$\sigma_{\mathrm{a,x}}(z) = \frac{c_\mathrm{f}\, \rho\, b\, I_\mathrm{v}(z_\mathrm{s})\, v_\mathrm{m}^2(z_\mathrm{s})}{m_{1,x}} R\, K_\mathrm{x}\, \Phi_{1,x}(z) \tag{9}$$

where

$c_\mathrm{f}$   is the force coefficient,
$\rho$   is the air density,
$b$   is the width of the structure,
$I_\mathrm{v}(z_\mathrm{s})$   is the turbulence intensity at the height $z = z_\mathrm{s}$ above ground,
$v_\mathrm{m}(z_\mathrm{s})$   is the mean wind velocity for $z = z_\mathrm{s}$,
$z_\mathrm{s}$   is the reference height,
$R$   is the square root of resonant response,
$K_\mathrm{x}$   is the non-dimensional coefficient,
$\phi_{1,x}(z)$   is the fundamental along-wind modal shape,
$m_{1,x}$   is the along-wind fundamental equivalent mass.

The following illustrates the detailed manual calculations for determining the peak acceleration at the top level of the studied tall CLT building in accordance to BS EN 1991-1-4, the corresponding UK NA, and ISO 10137.

The force coefficient $c_\mathrm{f}$ is equal to the net pressure coefficient given in the UK NA as 1.241. The air density in the UK is equal to 1.226 kg/m$^3$. The turbulence intensity at the height $z = z_\mathrm{s}$ above ground is equal to 0.134. Thus, the mean wind velocity $v_\mathrm{m}(z)$ at $z = z_\mathrm{s}$ can be calculated from Equation (4.3) of BS EN 1991-1-4 as

$$v_\mathrm{m}(z) = c_\mathrm{r}(z)\, c_\mathrm{o}(z)\, v_\mathrm{b} \tag{10}$$

Based on Figure NA.3 in the UK NA to BS EN 1991-1-4, the roughness factor $c_\mathrm{r}(z) = 1.37$ for $h_\mathrm{dis} = 0$ and a site distance of 40 km to the sea. In Clause NA.2.9, the orography factor $c_\mathrm{o}(z)$ is recommended as 1.0. The basic wind velocity $v_\mathrm{b}$ is determined as 26.21 m/s based on both BS EN 1991-1-4 and the UK NA. According to Equation (4.3) in BS EN 1991-1-4, the mean wind velocity $v_\mathrm{m}(z)$ is determined as $v_\mathrm{m}(z) = 1.37 \times 1.0 \times 26.21 = 35.91$ m/s.

According to Clause 4.2(2) Note 4 in BS EN 1991-1-4, the 10-min mean wind velocity having the probability $p$ for an annual exceedance is determined by multiplying the basic wind velocity $v_\mathrm{b}$ by the probability factor $c_\mathrm{prob}$, which is determined from

$$c_\mathrm{prob} = \left( \frac{1 - K \ln(-\ln(1-p))}{1 - K \ln(-\ln(0.98))} \right)^{\mathrm{n}} \tag{11}$$

Hence,

$$c_\mathrm{prob} = \left( \frac{1 - 0.2 \times \ln(-\ln(1-0.6321))}{1 - 0.2 \times \ln(-\ln(0.98))} \right)^{0.5} = 0.749$$

where the probability $p$ for an annual exceedance is reasonably assumed as $1 - 1/e = 0.6321$ [39], the shape parameter depending on the coefficient of variation of the extreme-value distribution, $K$, is recommended as 0.2, and the exponent index $n$ is recommended as 0.5. The mean wind velocity for $z = z_\mathrm{s}$ according to a one-year return period is equal to

$v_m(z) = 35.91 \times 0.749 = 26.90$ m/s. The resonance response factor $R$ can be obtained from Equation (B.6) in BS EN 1991-1-4:

$$R^2 = \frac{\pi^2}{2\delta} S_L(z_s, n_{1,x}) R_h(\eta_h) R_b(\eta_b) \tag{12}$$

According to Clause F.5 in BS EN 1991-1-4, the total logarithmic decrement of damping, $\delta$, for the fundamental bending mode may be estimated by using Equation (F.15) of the code as

$$\delta = \delta_s + \delta_a + \delta_d \tag{13}$$

where $\delta_s$ is the logarithmic decrement of structural damping, $\delta_a$ is the logarithmic decrement of aerodynamic damping for the fundamental mode, and $\delta_d$ is the logarithmic decrement of damping due to special devices (tuned mass dampers, sloshing tanks, etc.). Based on the listed result in Table F.2 in BS EN 1991-1-4, for timber bridges, $\delta_s$ could be chosen from 0.06 to 0.12. In this research, we assume $\delta_s = 0.06$ for timber buildings. $\delta_a$ can be obtained from Equation (F.18) in BS EN 1991-1-4 as

$$\delta_a = \frac{c_f \rho\, b\, v_m(z_s)}{2\, n_1\, m_e} \tag{14}$$

According to Clause F.4(2) in BS EN 1991-1-4, for cantilevered structures, the varying mass distribution $m_e$ may be approximated by the average value of $m$ over the upper third of the structure. For simplicity of the analysis, the average mass of the whole building is used for a varying mass distribution $m_e$ as $m_e = 236967$ kg/m. Thus,

$$\delta_a = \frac{1.241 \times 1.226 \times 39 \times 26.90}{2 \times 0.525 \times 236967} = 0.006$$

Based on Clause F.4(2), $\delta_d$ is assumed to be 0 here.

Hence, $\delta = 0.06 + 0.006 + 0 = 0.066$.

The non-dimensional power spectral density function $S_L(z, n)$ is given by Equation (B.2) in BS EN 1991-1-4 as

$$S_L(z,n) = \frac{n S_v(z,n)}{\sigma_v^2} = \frac{6.8 \times f_L(z,n)}{[1 + 10.2 \times f_L(z,n)]^{5/3}} \tag{15}$$

The non-dimensional frequency is determined by the frequency $n = n_{1,x}$ from Equation (B.2) in BS EN 1991-1-4 as

$$f_L(z,n) = \frac{n L(z)}{v_m(z)} \tag{16}$$

where the frequency $n = n_{1,x} = 0.525$ Hz. According to Equation (B.1), the turbulent length scale is

$$L(z_s) = \begin{cases} L_t \left(\dfrac{z}{z_t}\right)^\alpha & \text{for } z \geq z_{min} \\ L(z_{min}) & \text{for } z < z_{min} \end{cases} \tag{17}$$

$$\text{with} \alpha = 0.67 + 0.05 \ln(z_0) \tag{18}$$

Based on the data given in Table 4.1 of BS EN 1991-1-4, the roughness length $z_0 = 0.05$ m and the minimum height $z_{min} = 2$ m. Thus, $\alpha = 0.67 + 0.05 \times \ln(0.05) = 0.52$. For $z > z_{min}$, the reference height is $z_t = 200$ m, and the reference length scale $L_t = 300$ m. Thus, the turbulent length scale $L(z_s)$ can be obtained as

$$L(z_s) = L_t \left(\frac{z}{z_t}\right)^\alpha = 300 \times \left(\frac{63}{200}\right)^{0.52} = 164.49\, \text{m}.$$

The non-dimensional frequency $f_L(z, n)$ is determined as

$$f_L(z, n) = \frac{nL(z)}{v_m(z)} = \frac{0.525 \times 164.49}{26.90} = 3.211.$$

The non-dimensional power spectral density function $S_L(z, n)$ is

$$S_L(z, n) = \frac{6.8 \times 3.211}{(1 + 10.2 \times 3.211)^{5/3}} = 0.062.$$

The aerodynamic admittance functions about height, $R_h$, is given in Equation (B.7) in BS EN 1991-1-4 as

$$R_h = \frac{1}{\eta_h} - \frac{1}{2\eta_h^2}(1 - e^{-2\eta_h}) \tag{19}$$

$$\text{with}\, \eta_h = \frac{4.6\, h}{L(z_s)} f_L(z_s, n_{1,x}) \tag{20}$$

The value of the aerodynamic admittance function about height, $R_h$, is

$$R_h = \frac{1}{9.429} - \frac{1}{2 \times 9.429^2}(1 - e^{-2 \times 9.429}) = 0.1 \text{ with } \eta_h = \frac{4.6 \times 105}{164.49} \times 3.211 = 9.429.$$

The aerodynamic admittance function about width, $R_b$, is given in Equation (B.8) in BS EN 1991-1-4 as

$$R_b = \frac{1}{\eta_b} - \frac{1}{2\eta_b^2}(1 - e^{-2\eta_b}) \tag{21}$$

$$\text{with}\, \eta_b = \frac{4.6\, b}{L(z_s)} f_L(z_s, n_{1,x}) \tag{22}$$

Thus, the value of aerodynamic admittance functions about width, $R_b$, is

$$R_b = \frac{1}{3.502} - \frac{1}{2 \times 3.502^2}(1 - e^{-2 \times 3.502}) = 0.245 \text{ with } \eta_b = \frac{4.6 \times 39}{164.49} \times 3.211 = 3.502.$$

The resonance response factor $R$ is obtained as

$$R^2 = \frac{\pi^2}{2 \times 0.066} \times 0.062 \times 0.1 \times 0.245 = 0.113.$$

From Equation (F.13) in BS EN 1991-1-4, the fundamental along-wind modal shape is

$$\Phi_1(z) = \left(\frac{z}{h}\right)^\zeta \tag{23}$$

where, from Clause F.3 of the code, the parameter $\zeta = 1.0$ for buildings with a central core plus peripheral columns or larger columns plus shear bracings, and a building height ratio of $z_s/z_0 = 63/0.05 = 1260$. In Figure B.4 of the code, the non-dimensional coefficient $K_x = 1.5$. Thus, the fundamental along-wind modal shape $\phi_{1,x}(z) = 1$, because $z = 105$ m $= h$.

The standard deviation of the characteristic along-wind acceleration of the structural point at height $z$, $\sigma_a$, can then be obtained as

$$\sigma_a(z) = \frac{1.240 \times 1.226 \times 39 \times 0.134 \times 26.9^2}{236967} \times \sqrt{0.113} \times 1.5 \times 1 = 0.0122.$$

As mentioned in Equation (8), $k_p$ is a peak factor as, according to Equation (B.4) in BS EN 1991-1-4

$$k_p = \max \begin{cases} \sqrt{2\ln(vT)} + \dfrac{0.6}{\sqrt{2\ln(vT)}} \\ 3.0 \end{cases} \tag{24}$$

where, based on Clause B.4(4) in BS EN 1991-1-4, the natural frequency is assumed to be the up-crossing frequency, i.e., $v = n_{1,x} = 0.525$ Hz. Thus, the value of $k_p$ is

$$k_p = \max\left[\sqrt{2 \times \ln(0.525 \times 600)} + \frac{0.6}{\sqrt{2 \times \ln(0.525 \times 600)}}; 3.0\right] = 3.569.$$

Hence, the horizontal peak acceleration of the structure at height $z$, $A(z)$, is obtained as $A(z) = k_p \, \sigma_a(z) = 3.569 \times 0.0122 = 0.044$ m/s$^2$.

According to ISO 10137, the calculated horizontal peak acceleration is illustrated in Figure 8, together with the design limits for both residential and office buildings.

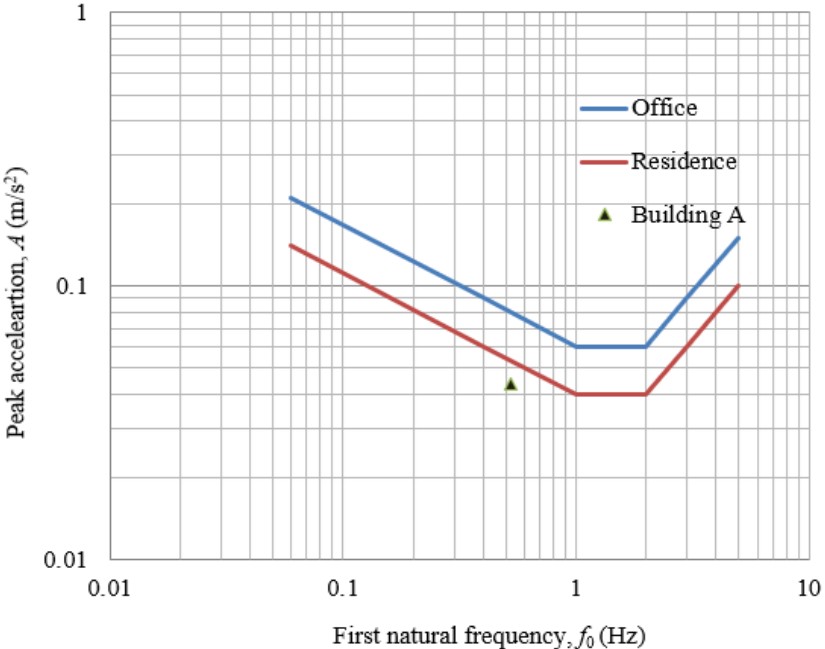

**Figure 8.** The calculated wind-induced peak acceleration with the evaluation curves for residential and office buildings along a horizontal (x or y) direction for a one-year return period.

In Figure 8, the point for the peak acceleration is below the limit line for office buildings, i.e., the blue line. Hence, the obtained acceleration at the top level of the studied building satisfies the requirement of ISO 10137.

*5.2. Parametric Studies on the Peak Accelerations of the Tall CLT Building*

To investigate the relationships between timber grade, mass, stiffness, and peak acceleration for the tall timber buildings, twelve different building models were established. Models A1 to A4 were used to assess the effects of timber grade on the peak acceleration, with their timber grades expanded from C24 for Model A to C16, C20, C30, and C35 for the others. The equivalent stiffness parameters of the CLT elements need to be re-calculated, with the details listed in Tables A1–A4 in Appendix A. Models A5 to A8 were used to assess the effects of the general building mass on the peak acceleration. Models A9 to A12 were used to assess the effects of the general building stiffness on the peak acceleration.

Many methods can be used to alter the mass and stiffness of the building, e.g., increasing the loadings on the CLT floor slabs or walls, increasing the vertical load-bearing elements, etc. To directly evaluate the relationships of the peak acceleration with the building mass and stiffness, Models A5 to A8 were assumed to vary the generalised building mass without altering other details, and Models A9 to A12 were assumed to vary the generalised building stiffness only without altering other details.

### 5.2.1. The Effect of Timber Grade on the Peak Acceleration of the Tall CLT Building

To assess the effect of timber grade on the peak acceleration of the tall CLT building, new building models were created by only changing the timber grade. Five different timber grades were adopted, including C16, C20, C24, C30, and C35. Model A1 uses C16 timber for the CLT structural elements, Model A2 uses C20 timber, Model A uses C24 timber, Model A3 uses C30 timber, and Model A4 uses C35 timber. Because of the change in the timber grade, the new equivalent properties of the different timber elements had to be recalculated, and the obtained properties are thus based on the estimation method. After running different numerical models using SAP2000, the obtained results are listed in Table 5.

**Table 5.** Parametric study on the peak acceleration by varying the timber material grade of the CLT members.

| Model | Timber Grade | Fundamental Frequency | Peak Acceleration |
|-------|-------------|----------------------|-------------------|
| A1 | C16 | 0.458 Hz | 0.053 m/s$^2$ |
| A2 | C20 | 0.493 Hz | 0.048 m/s$^2$ |
| A | C24 | 0.525 Hz | 0.044 m/s$^2$ |
| A3 | C30 | 0.540 Hz | 0.041 m/s$^2$ |
| A4 | C35 | 0.560 Hz | 0.039 m/s$^2$ |

Figure 9 illustrates the calculated peak accelerations at the top level of the building versus the fundamental frequency by varying the timber material grade from C16 to C35, together with the evaluation curves for residential and office buildings.

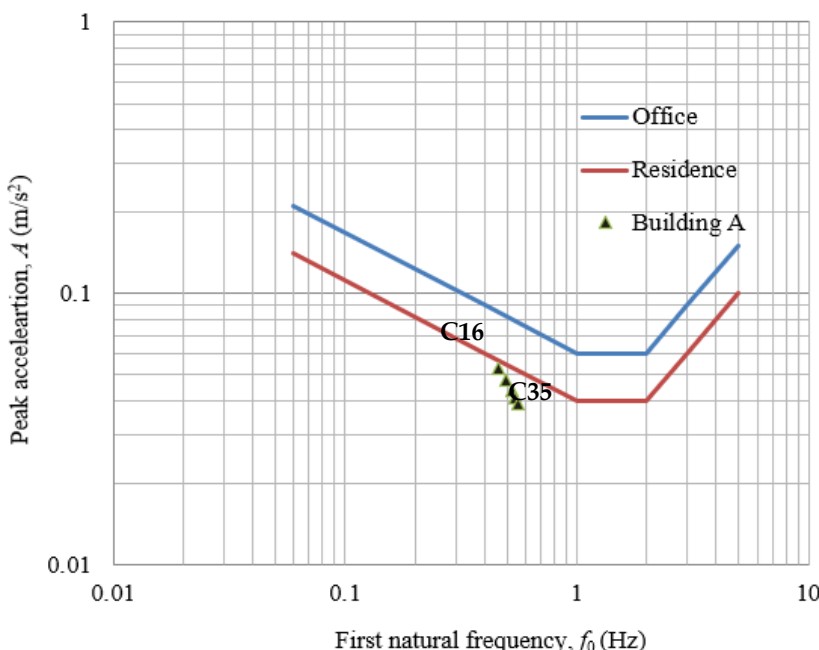

**Figure 9.** The calculated wind-induced peak accelerations with the evaluation curves for residential and office buildings along a horizontal (x or y) direction for a one-year return period by varying the timber grade.

As shown in Table 5 and Figure 9, with the increasing timber grade from C16 to C35, the fundamental vibrational frequency of the tall timber building increases and the acceleration at the top level of the building decreases from 0.053 m/s$^2$ to 0.039 m/s$^2$, down by 26.4%. Though the adopted timber grades are limited, the differences between the obtained results are obvious. This indicates that changing timber grade can be one practical option to help optimise the design of tall timber buildings.

### 5.2.2. The Effect of Building Mass on the Peak Acceleration of the Building

In order to study the effect of building mass on the peak acceleration of the building, four new models were created. Compared with Model A, Model A5 decreases the building mass by 40%, Model A6 decreases the building mass by 20%, Model A7 raises the building mass by 20%, and Model A8 raises the building mass by 40%.

According to the following general equation for determining the fundamental frequency and assuming the generalised stiffness of the building models has not changed, the fundamental frequency *f* can be directly calculated, and the results are listed in Table 6. Here, *K* is the generalised stiffness and *M* is the generalised mass for the building.

$$f = \frac{1}{2\pi}\sqrt{\frac{K}{M}} \tag{25}$$

**Table 6.** Parametric study on peak acceleration by varying the generalised building mass.

| Model | Building Mass | Fundamental Frequency | Peak Acceleration |
|-------|---------------|-----------------------|-------------------|
| A5 | −40% | 0.678 Hz | 0.054 m/s$^2$ |
| A6 | −20% | 0.587 Hz | 0.048 m/s$^2$ |
| A | 0% | 0.525 Hz | 0.044 m/s$^2$ |
| A7 | +20% | 0.479 Hz | 0.041 m/s$^2$ |
| A8 | +40% | 0.444 Hz | 0.038 m/s$^2$ |

Figure 10 illustrates the calculated peak accelerations at the top level of the building versus the fundamental frequency by varying the generalised building mass by up to ±40%, together with the evaluation curves for residential and office buildings.

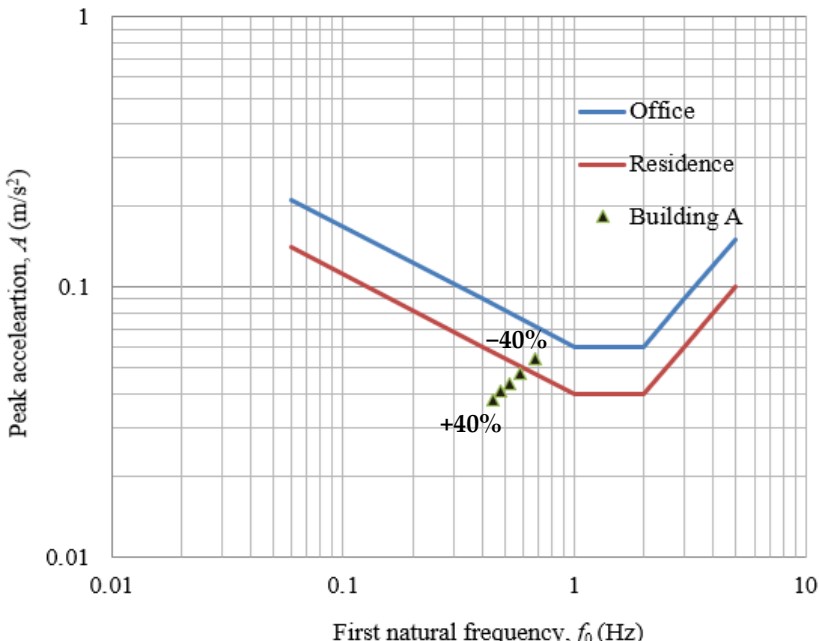

**Figure 10.** The calculated wind-induced peak accelerations with the evaluation curves for residential and office buildings along a horizontal (x or y) direction for a one-year return period by varying the building mass.

As shown in Table 6 and Figure 10, with the increasing building mass, the fundamental vibrational frequency of the tall timber building decreases and the acceleration at the top level of the building also decreases from 0.054 m/s$^2$ to 0.038 m/s$^2$, down by 29.6%. Though the adopted building masses are limited, the differences between the obtained results are

still obvious. This indicates that changing the building mass can be another practical option to help optimise the design of tall timber buildings.

### 5.2.3. The Effect of Building Stiffness on the Peak Acceleration of the Building

In order to study the effect of building stiffness on the peak acceleration of the building, four new models were created. Compared with Model A, Model A9 decreases the building stiffness by 40%, Model A10 decreases the building stiffness by 20%, Model A11 increases the building stiffness by 20%, and Model A12 is increases building stiffness by 40%. According to Equation (25), for determining the fundamental frequency and assuming the generalised building mass of the building models has not changed, the fundamental frequency $f$ can be calculated, and the results are listed in Table 7.

**Table 7.** Parametric study on peak acceleration by varying the generalised building stiffness.

| Model | Building Stiffness | Fundamental Frequency | Peak Acceleration |
|---|---|---|---|
| A9 | −40% | 0.407 Hz | 0.057 m/s$^2$ |
| A10 | −20% | 0.470 Hz | 0.049 m/s$^2$ |
| A | 0% | 0.525 Hz | 0.044 m/s$^2$ |
| A11 | +20% | 0.575 Hz | 0.040 m/s$^2$ |
| A12 | +40% | 0.621 Hz | 0.036 m/s$^2$ |

Figure 11 illustrates the calculated peak accelerations at the top level of the building versus the fundamental frequency by varying the generalised building stiffness by up to ±40%, together with the evaluation curves for residential and office buildings. As shown in Table 7 and Figure 11, with the increasing building stiffness, the fundamental vibrational frequency of the tall timber building increases but the acceleration at the top level of the building decreases from 0.057 m/s$^2$ to 0.036 m/s$^2$, down by 36.8%. Though the adopted building stiffnesses are limited, the differences between the obtained results still remain obvious. This indicates that changing the building stiffness can also be one practical option to help optimise the design of tall timber buildings.

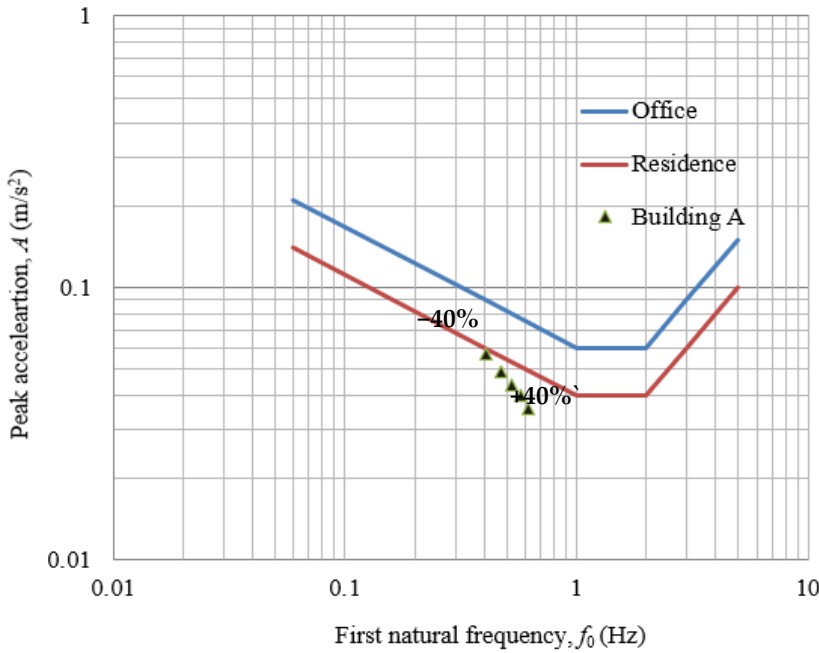

**Figure 11.** The calculated wind-induced peak accelerations with the evaluation curves for residential and office buildings along a horizontal (x or y) direction for a one-year return period by varying the building stiffness.

### 5.2.4. Discussion

In general, by increasing the timber grade, the generalised building mass and the generalised building stiffness can all be used to decrease the peak acceleration at the top level of the building, so as to reduce the human perception to the wind-induced vibrations with respect to peak acceleration. However, other criteria under ULS and SLS should be verified as well. Based on engineering judgement, if the effects of connections are considered, the vibration frequency and global stiffness of the building would decrease. Based on the results from Models A9 to A12, the values of the peak acceleration at the top level of the building would be increased accordingly.

### 6. Conclusions

A complete finite element building model was established and analysed for a 105 m-high 30-storey building constructed from pure CLT panels as the floor slabs, walls, and the lift core, based on the estimate method. The numerical simulations on the building model for the studied building illustrate the first two vibrational modes as translational modes and the third mode as a torsional mode, with the corresponding modal frequencies as 0.525 Hz, 0.579 Hz, and 0.678 Hz, and the first modal frequency is 19.9% larger than the code-recommended value of $46/105 = 0.438$ Hz. The fundamental frequency is almost the same as the estimated value suggested by Reynolds et al. At this moment, however, there are no restrictions on the fundamental frequency in the current structural Eurocodes, indicating that the obtained fundamental frequencies of this tall CLT building is reasonable.

Without considering the contributions of connections and shear deformations, the maximum horizontal displacement at the top level of the building is 28.9 mm, only 13.8% of the design limit to the current Eurocodes. The maximum static storey displacement for the building is 1.16 mm, only 16.6% of the design limit to the Eurocodes.

The horizontal peak acceleration at the top level of the studied building due to wind load was obtained and compared to the evaluation curves with respect to human vibration perception given in ISO 10137 for residential and office buildings along a horizontal direction for a one-year return period. The comparison indicates that the current peak acceleration is well below the limit line for office buildings, so the design is satisfactory.

A parametric study was also conducted to assess the peak acceleration at the top level of the building by varying the timber grade for CLT products, building mass, and building stiffness, and the comparisons confirm that, by increasing the timber grade, the generalised building mass and the generalised building stiffness can all be used to decrease the peak acceleration at the top level of the building.

This research is fairly sufficient and reliable for the design or assessment of the dynamic performance of a tall CLT building under wind action. So far, there are still no detailed real case studies that have been published regarding a dynamic analysis of tall CLT buildings under wind action. This research provides a thorough analysis method and procedure for civil engineers to assess the dynamic performance of tall CLT buildings under wind action. In the future, the effects of the connections between the CLT elements should be considered with the development of more powerful finite element software.

**Author Contributions:** Conceptualization, X.Z., B.Z., T.K. and I.S.; data curation, X.Z. and B.Z.; formal analysis, X.Z. and B.Z.; funding acquisition, B.Z. and T.K.; investigation, X.Z., B.Z. and T.K.; methodology, X.Z., B.Z. and T.K.; resources, T.K.; software, X.Z. and B.Z.; supervision, B.Z., T.K. and I.S.; validation, X.Z., B.Z., T.K. and I.S.; visualization, X.Z. and B.Z.; writing—original draft, X.Z. and B.Z.; writing—review and editing, X.Z., B.Z., T.K. and I.S. All authors have read and agreed to the published version of the manuscript.

**Funding:** This research received no external funding.

**Data Availability Statement:** The data presented in this research can be requested from the corresponding author or the first author.

**Acknowledgments:** This project is supported by the School of Computing, Engineering and Built Environment at Glasgow Caledonian University, Scotland, UK.

**Conflicts of Interest:** The authors declare no conflict of interest.

## Appendix A

**Table A1.** Equivalent stiffness parameters of the C16 CLT elements.

| Stiffness Parameters | Walls 1 and 2 | Wall 3 | Floor Slab |
|---|---|---|---|
| $t$ | 360.0 mm | 280.0 mm | 245.0 mm |
| $E_{rx}$ | 5412 N/mm$^2$ | 5768 N/mm$^2$ | 5767 N/mm$^2$ |
| $E_{ry}$ | 2858 N/mm$^2$ | 2501 N/mm$^2$ | 2500 N/mm$^2$ |
| $E_x$ | 3706 N/mm$^2$ | 3583 N/mm$^2$ | 3583 N/mm$^2$ |
| $E_y$ | 4564 N/mm$^2$ | 4687 N/mm$^2$ | 4687 N/mm$^2$ |
| $G_{xy}$ | 285.28 N/mm$^2$ | 292.95 N/mm$^2$ | 292.95 N/mm$^2$ |
| $G_{yz}$ | 80.81 N/mm$^2$ | 77.93 N/mm$^2$ | 77.92 N/mm$^2$ |
| $G_{zx}$ | 80.81 N/mm$^2$ | 77.93 N/mm$^2$ | 77.92 N/mm$^2$ |

**Table A2.** Equivalent stiffness parameters of the C20 CLT elements.

| Stiffness Parameters | Walls 1 and 2 | Wall 3 | Floor Slab |
|---|---|---|---|
| $t$ | 360.0 mm | 280.0 mm | 245.0 mm |
| $E_{rx}$ | 6427 N/mm$^2$ | 6851 N/mm$^2$ | 6847 N/mm$^2$ |
| $E_{ry}$ | 3392 N/mm$^2$ | 2970 N/mm$^2$ | 2968 N/mm$^2$ |
| $E_x$ | 4400 N/mm$^2$ | 4254 N/mm$^2$ | 4254 N/mm$^2$ |
| $E_y$ | 5420 N/mm$^2$ | 5566 N/mm$^2$ | 5566 N/mm$^2$ |
| $G_{xy}$ | 338.75 N/mm$^2$ | 347.86 N/mm$^2$ | 347.86 N/mm$^2$ |
| $G_{yz}$ | 95.36 N/mm$^2$ | 91.96 N/mm$^2$ | 91.96 N/mm$^2$ |
| $G_{zx}$ | 95.36 N/mm$^2$ | 91.96 N/mm$^2$ | 91.96 N/mm$^2$ |

**Table A3.** Equivalent stiffness parameters of the C30 CLT elements.

| Stiffness Parameters | Walls 1 and 2 | Wall 3 | Floor Slab |
|---|---|---|---|
| $t$ | 360.0 mm | 280.0 mm | 245.0 mm |
| $E_{rx}$ | 8117 N/mm$^2$ | 8655 N/mm$^2$ | 8646 N/mm$^2$ |
| $E_{ry}$ | 4282 N/mm$^2$ | 3749 N/mm$^2$ | 3746 N/mm$^2$ |
| $E_x$ | 5556 N/mm$^2$ | 5371 N/mm$^2$ | 5731 N/mm$^2$ |
| $E_y$ | 6844 N/mm$^2$ | 7029 N/mm$^2$ | 7029 N/mm$^2$ |
| $G_{xy}$ | 427.78 N/mm$^2$ | 439.29 N/mm$^2$ | 439.29 N/mm$^2$ |
| $G_{yz}$ | 121.22 N/mm$^2$ | 116.89 N/mm$^2$ | 116.90 N/mm$^2$ |
| $G_{zx}$ | 121.22 N/mm$^2$ | 116.89 N/mm$^2$ | 116.90 N/mm$^2$ |

**Table A4.** Equivalent stiffness parameters of the C35 CLT elements.

| Stiffness Parameters | Walls 1 and 2 | Wall 3 | Floor Slab |
|---|---|---|---|
| $t$ | 360.0 mm | 280.0 mm | 245.0 mm |
| $E_{rx}$ | 8794 N/mm$^2$ | 9371 N/mm$^2$ | 9372 N/mm$^2$ |
| $E_{ry}$ | 4637 N/mm$^2$ | 4059 N/mm$^2$ | 4056 N/mm2 |
| $E_x$ | 6017 N/mm$^2$ | 5817 N/mm$^2$ | 5817 N/mm$^2$ |
| $E_y$ | 7413 N/mm$^2$ | 7613 N/mm$^2$ | 7613 N/mm$^2$ |
| $G_{xy}$ | 463.33 N/mm$^2$ | 475.80 N/mm$^2$ | 475.80 N/mm$^2$ |
| $G_{yz}$ | 130.92 N/mm$^2$ | 126.25 N/mm$^2$ | 126.25 N/mm$^2$ |
| $G_{zx}$ | 130.92 N/mm$^2$ | 126.25 N/mm$^2$ | 126.25 N/mm$^2$ |

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
