# Peer review of "Numerical Analysis on Global Serviceability Behaviours of Tall CLT Buildings to the Eurocodes and UK National Annexes"

_buildings, doi:10.3390/buildings11030124_

Round 1

Reviewer 1 Report

Comments and Suggestions for Authors:

The article presents the static and dynamic FEM analyzes of CLT structure. Moreover, I have a few questions and comments:

1) The authors write that the models do not take into account the effects of connections in CLT, please explain?

2) please provide a detailed description of the CLT wall model in FEM, what is it made of? nodes, finite elements etc?

3) to what exactly is the alleged increase in acceleration peaks associated with the unaccounted for joining effect? please explain

4)What do authors mean when they write core made based on the estimate method. 

5) as the class of wood increases, the stiffness of the structure increases, this conclusion is obvious, did the authors manage to observe anything else? will this increase be proportional for each value?

6) what were the damping coefficients in dynamic analyzes and on what basis were they calculated?

7) the introduction requires improvement and expanded, some literature should be added. As a curiosity, a completely different approach to the problem of dynamic resistance of timber structures is indicated in the publications below, feel free to use them in introduction also, for instance:

https://doi.org/10.1061/(ASCE)ST.1943-541X.0000272

https://doi.org/10.3390/app9204387

From the editing point of view, it may be worth ordering the computational part of the article a bit, there are a lot of formulas in a slightly chaotic form. However, the article has considerable scientific value, especially since it concerns not very popular cross-laminated timber structures.

Author Response

Please see the attached file for our responses to the reviewer's comments and suggestions.

Reviewer 2 Report

The paper gives a contribution on the evaluation of design limits for tall buildings made of CLT panels.

It provides a significant case study, also useful for professionals; nevertheless, being submitted to a scientific journal, the manuscript needs some improvements, particularly in the methodology and justification of choices. Some suggestions are given in the following.

  1. Introduction:

- it is not clear why the detailed description of the Tallwood Hose is preferred to other buildings. Instead, the state-of-the-art would benefit from more detailed info to include for the cases listed in Table 1.

  1. Basic plans

- is the building under study a real building existing in Glasgow or is it a design project? Please specify.

- type of connectors among panel should be described/specified/illustrated (in case).Actually connectors play a fundamental role in buildings’ structural performance, therefore their  sentence ‘structural effects of actual connections are not considered here’ should be better justified.

  1. Finite element

- line 135: ‘it is necessary..’. FE is a method but not the only method suitable for these analyses. Please justify the adoption of FEM with respect to other methods/software. Some literature review is needed here.

- line 150: you do not need to report the journal of a reference or other stuff, but only the simple required reference format [number]. This also applyes to other parts of the paper, please check it.

- lines 154-156: please justify these assumptions/choices

- line 174-177: reference for Matsumoto and Yasumura? Please see previous comments.

- line 196: a comparison with not rigid connections would be in need (or at least better explored from literature review)

  1. Global performances

- line 227: what Model A means? Please clarify its properties/characteristics by better referring to the case study

- a table listing the dynamic parameters got for the main modes is missing; what about further modes besides the third one?

- line 243: which huge studies??

  1. Peak acceleration

- Models from 1 to 12 should be mentioned before, at least in their number when categories of investigation (CLT grade, mass, ..) are mentioned.

- line 478: ‘may be worse’, the sentence is too generic and needs to be justified with some data from literature, so that some ranges of variability can be given.

  1. Conclusions

- are the authors considering this study sufficient and reliable for design or assessment of tall buildings in CLT under wind actions?

Author Response

See the attached file for our responses to the comments and suggestions from the Reviewer.
